# Computational Separation Between Convolutional and Fully-Connected Networks

**Eran Malach**
School of Computer Science
Hebrew University
Jerusalem, Israel
eran.malach@mail.huji.ac.il

**Shai Shalev-Shwartz**
School of Computer Science
Hebrew University
Jerusalem, Israel
shais@cs.huji.ac.il

## Abstract

Convolutional neural networks (CNN) exhibit unmatched performance in a multitude of computer vision tasks. However, the advantage of using convolutional networks over fully-connected networks is not understood from a theoretical perspective. In this work, we show how convolutional networks can leverage locality in the data, and thus achieve a computational advantage over fully-connected networks. Specifically, we show a class of problems that can be efficiently solved using convolutional networks trained with gradient-descent, but at the same time is hard to learn using a polynomial-size fully-connected network.

## 1 Introduction

Convolutional neural networks (LeCun et al., 1998; Krizhevsky et al., 2012) achieve state-of-the-art performance on every possible task in computer vision. However, while the empirical success of convolutional networks is indisputable, the advantage of using them is not well understood from a theoretical perspective. Specifically, we consider the following fundamental question:

*Why do convolutional networks (CNNs) perform better than fully-connected networks (FCNs)?*

Clearly, when considering expressive power, FCNs have a big advantage. Since convolution is a linear operation, any CNN can be expressed using a FCN, whereas FCNs can express a strictly larger family of functions. So, any advantage of CNNs due to expressivity can be leveraged by FCNs as well. Therefore, *expressive power does not explain the superiority of CNNs over FCNs*.

There are several possible explanations to the superiority of CNNs over FCNs: parameter efficiency (and hence lower sample complexity), weight sharing, and locality prior. The main result of this paper is arguing that locality is a key factor by proving a computational separation between CNNs and FCNs based on locality. But, before that, let's discuss the other possible explanations.

First, we observe that CNNs seem to be much more efficient in utilizing their parameters. A FCN needs to use a greater number of parameters compared to an equivalent CNN: each neuron of a CNN is limited to a small receptive field, and moreover, many of the parameters of the CNN are shared. From classical results in learning theory, using a large number of parameters may result in inferior generalization. So, can the advantage of CNNs be explained simply by counting parameters?

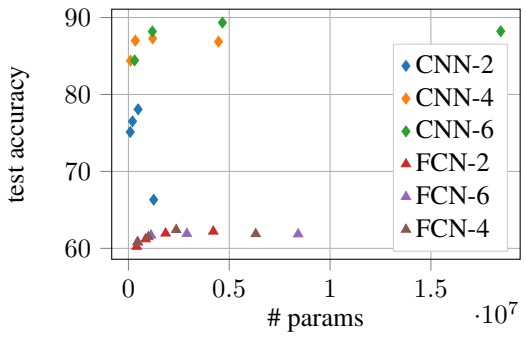

Figure 1: Comparison between CNN and FCN of various depths (2/4/6) and widths, trained for 125 epochs with RMSprop optimizer.

To answer this question, we observe the performance of CNN and FCN based architecture of various widths and depths trained on the CIFAR-10 dataset. For each architecture, we observe the final test accuracy versus the number of trainable parameters. The results are shown in Figure 1. As can be seen, CNNs have a clear advantage over FCNs, regardless of the number of parameters used. As is often observed, a large number of parameters does not hurt the performance of neural networks, and so *parameter efficiency cannot explain the advantage of CNNs*. This is in line with various theoretical works on optimization of neural networks, which show that over-parameterization is beneficial for convergence of gradient-descent (e.g., Du et al. (2018); Soltanolkotabi et al. (2018); Li & Liang (2018)).

The superiority of CNNs can be also attributed to the extensive *weight sharing* between the different convolutional filters. Indeed, it has been previously shown that weight sharing is important for the optimization of neural networks (Shalev-Shwartz et al., 2017b). Moreover, the translation-invariant nature of CNNs, which relies on weight sharing, is often observed to be beneficial in various signal processing tasks (Kauderer-Abrams, 2017; Kayhan & Gemert, 2020). So, how much does the weight sharing contribute to the superiority of CNNs over FCNs?

To understand the effect of weight sharing on the behavior of CNNs, it is useful to study **locally-connected network** (LCN) architectures, which are similar to CNNs, but have no weight sharing between the kernels of the network. While CNNs are far more popular in practice (also due to the fact that they are much more efficient in terms of model size), LCNs have also been used in different contexts (e.g., Bruna et al. (2013); Chen et al. (2015); Liu et al. (2020)). It has been recently observed that in some cases, the performance of LCNs is on par with CNNs (Neyshabur, 2020). So, even if weight sharing explains some of the advantage of CNNs, it clearly doesn't tell the whole story.

Finally, a key property of CNN architectures is their strong utilization of **locality** in the data. Each neuron in a CNN is limited to a local receptive field of the input, hence encoding a strong locality bias. In this work we demonstrate how CNNs can leverage the local structure of the input, giving them a clear advantage in terms of *computational complexity*. Our results hint that **locality** is the principal property that explains the advantage of using CNNs.

Our main result is a computational separation result between CNNs and FCNs. To show this result, we introduce a family of functions that have a very strong local structure, which we call $k$-**patterns**. A $k$-**pattern** is a function that is determined by $k$ *consecutive* bits of the input. We show that for inputs of $n$ bits, when the target function is a $(\log n)$-pattern, training a CNN of polynomial size with gradient-descent achieves small error in polynomial time. However, gradient-descent will fail to learn $(\log n)$-patterns, when training a FCN of polynomial-size.

## 1.1 RELATED WORK

It has been empirically observed that CNN architectures perform much better than FCNs on computer vision tasks, such as digit recognition and image classification (e.g., Urban et al. (2017); Driss et al. (2017)). While some works have applied various techniques to improve the performance of FCNs (Lin et al. (2015); Fernando et al. (2016); Neyshabur (2020)), there is still a gap between performance of CNNs and FCNs, where the former give very good performance "out-of-the-box". The focus of this work is to understand, from a theoretical perspective, why CNNs give superior performance when trained on input with strong local structure.

Various theoretical works show the advantage of architectures that leverage local and hierarchical structure. The work of Poggio et al. (2015) shows the advantage of using deep hierarchical models over wide and shallow functions. These results are extended in Poggio et al. (2017), showing an exponential gap between deep and shallow networks, when approximating locally compositional functions. The works of Mossel (2016); Malach & Shalev-Shwartz (2018) study learnability of deep hierarchical models. The work of Cohen et al. (2017) analyzes the expressive efficiency of convolutional networks via hierarchical tensor decomposition. While all these works show that indeed CNNs powerful due to their hierarchical nature and the efficiency of utilizing local structure, they do not explain why these models are superior to fully-connected models.

There are a few works that provide a theoretical analysis of CNN optimization. The works of Brutzkus & Globerson (2017); Du et al. (2018) show that gradient-descent can learn a shallow CNN with a single filter, under various distributional assumptions. The work of Zhang et al. (2017)

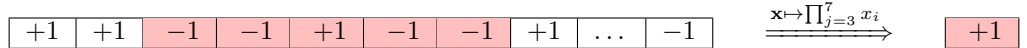

Figure 2: Example of a $k$-pattern with $k = 5$.

shows learnability of a convex relaxation of convolutional networks. While these works focus on computational properties of learning CNNs, as we do in this work, they do not compare CNNs to FCNs, but focus only on the behavior of CNNs. The works of Cohen & Shashua (2016); Novak et al. (2018) study the implicit bias of simplified CNN models. However, these result are focused on generalization properties of CNNs, and not on computational efficiency of the optimization.

## 2  DEFINITIONS AND NOTATIONS

Let $\mathcal{X} = \{\pm 1\}^n$ be our instance space, and let $\mathcal{Y} = \{\pm 1\}$ be the label space. Throughout the paper, we focus on learning a binary classification problem using the hinge-loss: $\ell(\hat{y}, y) = \max\{1 - y\hat{y}, 0\}$. Given some distribution $\mathcal{D}$ over $\mathcal{X}$, some target function $f : \mathcal{X} \to \mathcal{Y}$ and some hypothesis $h : \mathcal{X} \to \mathcal{Y}$, we define the loss of $h$ with respect to $f$ on the distribution $\mathcal{D}$ by:

$$L_{f,\mathcal{D}}(h) = \mathop{\mathbb{E}}_{\mathbf{x} \sim \mathcal{D}} [\ell(h(\mathbf{x}), f(\mathbf{x}))]$$

The goal of a supervised learning algorithm is, given access to examples sampled from $\mathcal{D}$ and labeled by $f$, to find a hypothesis $h$ that minimizes $L_{f,\mathcal{D}}(h)$. We focus on the gradient-descent (GD) algorithm: given some parametric hypothesis class $\mathcal{H} = \{h_{\mathbf{w}} : \mathbf{w} \in \mathbb{R}^q\}$, gradient-descent starts with some (randomly initialized) hypothesis $h_{\mathbf{w}^{(0)}}$ and, for some learning rate $\eta > 0$, updates:

$$\mathbf{w}^{(t)} = \mathbf{w}^{(t-1)} - \eta \nabla_{\mathbf{w}} L_{f,\mathcal{D}}(h_{\mathbf{w}^{(t-1)}})$$

We compare the behavior of gradient-descent, when learning two possible neural network architectures: a **convolutional network** (CNN) and a **fully-connected network** (FCN).

**Definition 1.** *A **convolutional network** $h_{\mathbf{u},W,\mathbf{b}}$ is defined as follows:*

$$h_{\mathbf{u},W,b}(\mathbf{x}) = \sum_{j=1}^{n-k} \left\langle \mathbf{u}^{(j)}, \sigma(W \mathbf{x}_{j \dots j+k-1} + \mathbf{b}) \right\rangle$$

*for activation function $\sigma$, with kernel $W \in \mathbb{R}^{q \times k}$, bias $\mathbf{b} \in \mathbb{R}^q$ and readout layer $\mathbf{u}^{(1)}, \dots, \mathbf{u}^{(n)} \in \mathbb{R}^q$. Note that this is a standard depth-2 CNN with kernel $k$, stride $1$ and $q$ filters.*

**Definition 2.** *A **fully-connected network** $h_{\mathbf{u},\mathbf{w},\mathbf{b}}$ is defined as follows:*

$$h_{\mathbf{u},\mathbf{w},\mathbf{b}}(\mathbf{x}) = \sum_{i=1}^{q} u_i \sigma \left( \left\langle \mathbf{w}^{(i)}, \mathbf{x} \right\rangle + b_i \right)$$

*for activation function $\sigma$, first layer $\mathbf{w}^{(1)}, \dots, \mathbf{w}^{(q)} \in \mathbb{R}^n$, bias $\mathbf{b} \in \mathbb{R}^q$ and second layer $\mathbf{u} \in \mathbb{R}^q$.*

We demonstrate the advantage of CNNs over FCNs by observing a problem that can be learned using CNNs, but is hard to learn using FCNs. We call this problem the $k$-**pattern** problem:

**Definition 3.** *A function $f : \mathcal{X} \to \mathcal{Y}$ is a $k$-**pattern**, if for some $g : \{\pm 1\}^k \to \mathcal{Y}$ and index $j^*$:*

$$f(\mathbf{x}) = g(x_{j^* \dots j^* + k - 1})$$

Namely, a $k$-**pattern** is a function that depends only on a small pattern of *consecutive* bits of the input. The $k$-**pattern problem** is the problem of learning $k$-patterns: for some $k$-pattern $f$ and some distribution $\mathcal{D}$ over $\mathcal{X}$, given access to $\mathcal{D}$ labeled by $f$, find a hypothesis $h$ with $L_{f,\mathcal{D}}(h) \leq \epsilon$. We note that a similar problem has been studied in Golovnev et al. (2017), providing results on PAC learnability of a related target class.

## 3 CNNs Efficiently Learn $(\log n)$-Patterns

The main result in this section shows that gradient-descent can learn $k$-patterns when training convolutional networks for $poly(2^k, n)$ iterations, and when the network has $poly(2^k, n)$ neurons:

**Theorem 4.** *Assume we uniformly initialize $W^{(0)} \sim \{\pm 1/k\}^{q \times k}$, $b_i = 1/k - 1$ and $\mathbf{u}^{(0,j)} = 0$ for every $j$. Assume the activation $\sigma$ satisfies $|\sigma| \leq c, |\sigma'| \leq 1$, for some constant $c$. Fix some $\delta > 0$, some $k$-pattern $f$ and some distribution $\mathcal{D}$ over $\mathcal{X}$. Then, if $q > 2^{k+3} \log(2^k/\delta)$, with probability at least $1 - \delta$ over the initialization, when training a convolutional network $h_{\mathbf{u}, W, \mathbf{b}}$ using gradient descent with $\eta = \frac{\sqrt{n}}{\sqrt{q}T}$ we have:*

$$\frac{1}{T} \sum_{t=1}^{T} L_{f,\mathcal{D}}(h_{\mathbf{u}^{(t)}, W^{(t)}, b}) \leq \frac{2cn^2 k^2 2^k}{q} + \frac{2(2^k k)^2}{\sqrt{qn}} + \frac{c^2 n^{1.5} \sqrt{q}}{T}$$

Before we prove the theorem, observe that the above immediately implies that when $k = O(\log n)$, gradient-descent can **efficiently** learn to solve the $k$-pattern problem, when training a CNN:

**Corollary 5.** *Let $k = O(\log n)$. Then, running GD on a CNN with $q = O(\epsilon^{-2} n^3 \log^2 n)$ neurons for $T = O(\epsilon^{-2} n^3 \log n)$ iterations, using a sample $S \sim \mathcal{D}$ of size $O(\epsilon^{-2} nkq \log(nkq/\delta))$, learns the $k$-pattern problem up to accuracy $\epsilon$ w.p. $\geq 1 - \delta$.*

*Proof.* Sample $S \sim \mathcal{D}$, and let $\widehat{\mathcal{D}}$ be the uniform distribution over $S$. Then, from Theorem 4 and the choice of $q$ and $T$ there exists $t \in [T]$ with $L_{f, \widehat{\mathcal{D}}}(h_{\mathbf{u}^{(t)}, W^{(t)}, b}) \leq \epsilon/2$, i.e. GD finds a hypothesis with train loss at most $\epsilon/2$. Now, using the fact the VC dimension of depth-2 ReLU networks with $W$ weights is $O(W \log W)$ (see Bartlett et al. (2019)), we can bound the generalization gap by $\epsilon/2$. □

To prove Theorem 4, we show that, for a large enough CNN, the $k$-pattern problem becomes linearly separable, after applying the first layer of the randomly initialized CNN:

**Lemma 6.** *Assume we uniformly initialize $W \sim \{\pm 1/k\}^{q \times k}$ and $b_i = 1/k - 1$. Fix some $\delta > 0$. Then if $q > 2^{k+3} \log(2^k/\delta)$, w.p. $\geq 1 - \delta$ over the choice of $W$, for every $k$-pattern $f$ there exist $\mathbf{u}^{*(1)}, \ldots, \mathbf{u}^{*(n-k)} \in \mathbb{R}^q$ with $\left\| \mathbf{u}^{*(j^*)} \right\| \leq \frac{2^{k+1}k}{\sqrt{q}}$ and $\left\| \mathbf{u}^{*(j)} \right\| = 0$ for $j \neq j^*$, s.t. $h_{\mathbf{u}^*, W, b} = f(\mathbf{x})$.*

*Proof.* Fix some $\mathbf{z} \in \{\pm 1\}^k$, then for every $\mathbf{w}^{(i)} \sim \{\pm 1/k\}^k$, we have: $\mathbb{P}\left[\text{sign}(\mathbf{w}^{(i)}) = \mathbf{z}\right] = 2^{-k}$. Denote by $J_{\mathbf{z}} \subseteq [q]$ the subset of indexes satisfying $\text{sign } \mathbf{w}^{(i)} = \mathbf{z}$, for every $i \in J_{\mathbf{z}}$, and note that $\mathbb{E}_W |J_{\mathbf{z}}| = q2^{-k}$. From Chernoff bound:

$$\mathbb{P}\left[|J_{\mathbf{z}}| \leq q2^{-k}/2\right] \leq e^{-q2^{-k}/8} \leq \delta 2^{-k}$$

by choosing $q > 2^{k+3} \log(2^k/\delta)$. So, using the union bound, w.p. at least $1 - \delta$, for every $\mathbf{z} \in \{\pm 1\}^k$ we have $|J_{\mathbf{z}}| \geq q2^{-k-1}$. By the choice of $b_i$ we have $\sigma(\langle \mathbf{w}^{(i)}, \mathbf{z} \rangle + b_i) = (1/k)\mathbf{1}\{\text{sign } \mathbf{w}^{(i)} = \mathbf{z}\}$.

Now, fix some $k$-pattern $f$, where $f(\mathbf{x}) = g(\mathbf{x}_{j^*, \ldots, j^*+k-1})$. For every $i \in J_{\mathbf{z}}$ we choose $\mathbf{u}_i^{*(j^*)} = \frac{k}{|J_{\mathbf{z}}|} g(\mathbf{z})$ and $\mathbf{u}^{*(j)} = 0$ for every $j \neq j^*$. Therefore, we get:

$$h_{\mathbf{u}^*, b}(\mathbf{x}) = \sum_{j=1}^{n-k} \left\langle \mathbf{u}^{*(j)}, \sigma(W\mathbf{x}_{j \ldots j+k-1} + \mathbf{b}) \right\rangle = \sum_{\substack{\mathbf{z} \in \{\pm 1\}^k \\ i \in J_{\mathbf{z}}}} \mathbf{u}_i^{*(j^*)} \sigma\left(\left\langle \mathbf{w}^{(i)}, \mathbf{x}_{j^* \ldots j^*+k-1} \right\rangle + b_i\right)$$

$$= \sum_{\mathbf{z} \in \{\pm 1\}^k} \mathbf{1}\{\mathbf{z} = \mathbf{x}_{j^* \ldots j^*+k-1}\} g(\mathbf{z}) = g(x_{j^* \ldots j^*+k-1}) = f(\mathbf{x})$$

Note that by definition of $\mathbf{u}^{*(j^*)}$ we have $\left\| \mathbf{u}^{*(j^*)} \right\|^2 = \sum_{\mathbf{z} \in \{\pm 1\}^k} \sum_{i \in J_{\mathbf{z}}} \frac{k^2}{|J_{\mathbf{z}}|^2} \leq 4 \frac{(2^k k)^2}{q}$. □

**Comment 7.** Admittedly, the initialization assumed above is non-standard, but is favorable for the analysis. A similar result can be shown for more natural initialization (e.g., normal distribution), using known results from random features analysis (for example, Bresler & Nagaraj (2020)).

From Lemma 6 and known results on learning linear classifiers with gradient-descent, solving the $k$-pattern problem can be achieved by optimizing the second layer of a randomly initialized CNN. However, since in gradient-descent we optimize both layers of the network, we need a more refined analysis to show that full gradient-descent learns to solve the problem. We follow the scheme introduced in Daniely (2017), adapting it our setting.

We start by showing that the first layer of the network does not deviate from the initialization during the training:

**Lemma 8.** *We have* $\left\|\mathbf{u}^{(T,j)}\right\| \le \eta T \sqrt{q}$ *for all* $j \in [n-k]$*, and* $\left\|W^{(0)} - W^{(T)}\right\| \le c\eta^2 T^2 n\sqrt{qk}$

We can now bound the difference in the loss when the weights of the first layer change during the training process:

**Lemma 9.** *For every* $\mathbf{u}^*$ *we have:*

$$\left| L_{f,\mathcal{D}}(h_{\mathbf{u}^*,W^{(T)},b}) - L_{f,\mathcal{D}}(h_{\mathbf{u}^*,W^{(0)},b}) \right| \le c\eta^2 T^2 nk\sqrt{q} \sum_{j=1}^{n-k} \left\|\mathbf{u}^{*(j)}\right\|$$

The proofs of Lemma 8 and Lemma 9 are shown in the appendix.

Finally, we use the following result on the convergence of online gradient-descent to show that gradient-descent converges to a good solution. The proof of the Theorem is given in Shalev-Shwartz et al. (2011), with an adaptation to a similar setting in Daniely & Malach (2020).

**Theorem 10.** *(Online Gradient Descent) Fix some* $\eta$*, and let* $f_1, \ldots, f_T$ *be some sequence of convex functions. Fix some* $\theta_1$*, and update* $\theta_{t+1} = \theta_t - \eta \nabla f_t(\theta_t)$*. Then for every* $\theta^*$ *the following holds:*

$$\frac{1}{T} \sum_{t=1}^{T} f_t(\theta_t) \le \frac{1}{T} \sum_{t=1}^{T} f_t(\theta^*) + \frac{1}{2\eta T} \|\theta^*\|^2 + \|\theta_1\| \frac{1}{T} \sum_{t=1}^{T} \|\nabla f_t(\theta_t)\| + \eta \frac{1}{T} \sum_{t=1}^{T} \|\nabla f_t(\theta_t)\|^2$$

*Proof of Theorem 4.* From Lemma 6, with probability at least $1 - \delta$ over the initialization, there exist $\mathbf{u}^{*(1)}, \ldots, \mathbf{u}^{*(n-k)} \in \mathbb{R}^q$ with $\left\|\mathbf{u}^{*(1)}\right\| \le \frac{2^{k+1}k}{\sqrt{q}}$ and $\left\|\mathbf{u}^{*(j)}\right\| = 0$ for $j > 1$ such that $h_{\mathbf{u}^*,W^{(0)},b}(\mathbf{x}) = f(\mathbf{x})$, and so $L_{f,\mathcal{D}}(h_{\mathbf{u}^*,W^{(0)},b}) = 0$. Using Theorem 10, since $L_{f,\mathcal{D}}(h_{\mathbf{u},W,b})$ is convex with respect to $\mathbf{u}$, we have:

$$\frac{1}{T} \sum_{t=1}^{T} L_{f,\mathcal{D}}(h_{\mathbf{u}^{(t)},W^{(t)},b})$$

$$\le \frac{1}{T} \sum_{t=1}^{T} L_{f,\mathcal{D}}(h_{\mathbf{u}^*,W^{(t)},b}) + \frac{1}{2\eta T} \sum_{j=1}^{n-k} \left\|\mathbf{u}^{*(j)}\right\|^2 + \eta \frac{1}{T} \sum_{t=1}^{T} \left\|\frac{\partial}{\partial \mathbf{u}} L_{f,\mathcal{D}}(f_{\mathbf{u}^{(t)},W^{(t)},b})\right\|^2$$

$$\le \frac{1}{T} \sum_{t=1}^{T} L_{f,\mathcal{D}}(h_{\mathbf{u}^*,W^{(t)},b}) + \frac{2(2^k k)^2}{q\eta T} + c^2 \eta n q = (*)$$

Using Lemma 9 we have:

$$(*) \le \frac{1}{T} \sum_{t=1}^{T} L_{f,\mathcal{D}}(h_{\mathbf{u}^*,W^{(0)},b}) + c\eta^2 T^2 nk\sqrt{q} \sum_{j=1}^{n-k} \left\|\mathbf{u}^{*(j)}\right\| + \frac{2(2^k k)^2}{q\eta T} + c^2 \eta n q$$

$$\le 2c\eta^2 T^2 nk^2 2^k + \frac{2(2^k k)^2}{q\eta T} + c^2 \eta n q$$

Now, choosing $\eta = \frac{\sqrt{n}}{\sqrt{qT}}$ we get the required. $\qquad\square$

## 3.1 Analysis of Locally-Connected Networks

The above result shows that polynomial-size CNNs can learn $(\log n)$-patterns in polynomial time. As discussed in the introduction, the success of CNNs can be attributed to either the weight sharing

or the locality-bias of the architecture. While weight sharing may contribute to the success of CNNs in some cases, we note that it gives no benefit when learning $k$-patterns. Indeed, we can show a similar positive result for locally-connected networks (LCN), which have no weight sharing.

Observe the following definition of a LCN with one hidden-layer:

**Definition 11.** *A **locally-connected network** $h_{\mathbf{u},\mathbf{w},\mathbf{b}}$ is defined as follows:*

$$h_{\mathbf{u},\mathbf{W},\mathbf{b}}(\mathbf{x}) = \sum_{j=1}^{n-k} \left\langle \mathbf{u}^{(j)}, \sigma(W^{(j)}\mathbf{x}_{j\ldots j+k-1} + \mathbf{b}^{(j)}) \right\rangle$$

*for some activation function $\sigma$, with $W^{(1)}, \ldots, W^{(q)} \in \mathbb{R}^{q \times k}$, bias $\mathbf{b}^{(1)}, \ldots, \mathbf{b}^{(q)} \in \mathbb{R}^q$ and readout layer $\mathbf{u}^{(1)}, \ldots, \mathbf{u}^{(n)} \in \mathbb{R}^q$.*

Note that the only difference from Definition 1 is the fact that the weights of the first layer are not shared. It is easy to verify that Theorem 4 can be modified in order to show a similar positive result for LCN architectures. Specifically, we note that in Lemma 6, which is the core of the Theorem, we do not use the fact that the weights in the first layer are shared. So, LCNs are "as good as" CNNs for solving the $k$-pattern problem. This of course does not resolve the question of comparing between LCN and CNN architectures, which we leave for future work.

## 4 LEARNING $(\log n)$-PATTERNS WITH FCN

In the previous section we showed that patterns of size $\log n$ are efficiently learnable, when using CNNs trained with gradient-descent. In this section we show that, in contrast, gradient-descent fails to learn $(\log n)$-patterns using fully-connected networks, unless the size of the network is super-polynomial (namely, unless the network is of size $n^{\Omega(\log n)}$). For this, we will show an instance of the $k$-pattern problem that is hard for fully connected networks.

We take $\mathcal{D}$ to be the uniform distribution over $\mathcal{X}$, and let $f(\mathbf{x}) = \prod_{i \in I} x_i$, where $I$ is some set of $k$ consecutive bits. Specifically, we take $I = \{1, \ldots, k\}$, although the same proof holds for any choice of $I$. In this case, we show that the initial gradient of the network is very small, when a fully-connected network is initialized from a permutation invariant distribution.

**Theorem 12.** *Assume $|\sigma| \leq c, |\sigma'| \leq 1$. Let $\mathcal{W}$ be some permutation invariant distribution over $\mathbb{R}^n$, and assume we initialize $\mathbf{w}^{(1)}, \ldots, \mathbf{w}^{(q)} \sim \mathcal{W}$ and initialize $\mathbf{u}$ such that $|u_i| \leq 1$ and for all $\mathbf{x}$ we have $h_{\mathbf{u},\mathbf{w}}(\mathbf{x}) \in [-1, 1]$. Then, the following holds:*

- $\mathbb{E}_{\mathbf{w} \sim \mathcal{W}} \left\| \frac{\partial}{\partial W} L_{f,\mathcal{D}}(h_{\mathbf{u},\mathbf{w},\mathbf{b}}) \right\|_2^2 \leq qn \cdot \min \left\{ \binom{n-1}{k}^{-1}, \binom{n-1}{k-1}^{-1} \right\}$

- $\mathbb{E}_{\mathbf{w} \sim \mathcal{W}} \left\| \frac{\partial}{\partial \mathbf{u}} L_{f,\mathcal{D}}(h_{\mathbf{u},\mathbf{w},\mathbf{b}}) \right\|_2^2 \leq c^2 q \binom{n}{k}^{-1}$

From the above result, if $k = \Omega(\log n)$ then the average norm of initial gradient is $qn^{-\Omega(\log n)}$. Therefore, unless $q = n^{\Omega(\log n)}$, we get that with overwhelming probability over the randomness of the initialization, the gradient is extremely small. In fact, if we run GD on a finite-precision machine, the true population gradient is effectively zero. A formal argument relating such bound on the gradient norm to the failure of gradient-based algorithms has been shown in various previous works (e.g. Shamir (2018); Abbe & Sandon (2018); Malach & Shalev-Shwartz (2020)).

The key for proving Theorem 12 is the following observation: since the first layer of the FCN is initialized from a symmetric distribution, we observe that if learning *some* function that relies on $k$ bits of the input is hard, then learning *any* function that relies on $k$ bits is hard. Using Fourier analysis (e.g., Blum et al. (1994); Kearns (1998); Shalev-Shwartz et al. (2017a)), we can show that learning $k$-parities (functions of the form $\mathbf{x} \mapsto \prod_{i \in I} x_i$) using gradient-descent is hard. Since an arbitrary $k$-parity is hard, then any $k$-parity, and specifically a parity of $k$ consecutive bits, is also hard. That is, since the first layer is initialized symmetrically, training a FCN on the original input is equivalent to training a FCN on an input where all the input bits are randomly permuted. So, for a FCN, learning a function that depends on consecutive bits is just as hard as learning a function that depends on arbitrary bits (a task that is known to be hard).

*Proof of Theorem 12.* Denote $\chi_{I'} = \prod_{i \in I'} x_i$, so $f(\mathbf{x}) = \chi_I$ with $I = \{1, \ldots, k\}$. We begin by calculating the gradient w.r.p. to $\mathbf{w}_j^{(i)}$:

$$\frac{\partial}{\partial \mathbf{w}_j^{(i)}} L_{f,\mathcal{D}}(h_{\mathbf{u},\mathbf{w},\mathbf{b}}) = \mathbb{E}_{\mathcal{D}} \left[ \frac{\partial}{\partial \mathbf{w}_j^{(i)}} \ell(h_{\mathbf{u},\mathbf{w},\mathbf{b}}(\mathbf{x}), f(\mathbf{x})) \right] = -\mathbb{E}_{\mathcal{D}} \left[ x_j u_i \sigma' \left( \left\langle \mathbf{w}^{(i)}, \mathbf{x} \right\rangle + b_i \right) \chi_I(\mathbf{x}) \right]$$

Fix some permutation $\pi : [n] \to [n]$. For some vector $\mathbf{x} \in \mathbb{R}^n$ we denote $\pi(\mathbf{x}) = (x_{\pi(1)}, \ldots, x_{\pi(n)})$, for some subset $I \subseteq [n]$ we denote $\pi(I) = \cup_{j \in I} \{\pi(j)\}$. Notice that we have for all $\mathbf{x}, \mathbf{z} \in \mathbb{R}^n$: $\chi_I(\pi(\mathbf{x})) = \chi_{\pi(I)}$ and $\langle \pi(\mathbf{x}), \mathbf{z} \rangle = \langle \mathbf{x}, \pi^{-1}(\mathbf{z}) \rangle$. Denote $\pi(h_{\mathbf{u},\mathbf{w},\mathbf{b}})(\mathbf{x}) = \sum_{i=1}^k u_i \sigma(\langle \pi(\mathbf{w}^{(i)}), \mathbf{x} \rangle + b_i)$. Denote $\pi(\mathcal{D})$ the distribution of $\pi(\mathbf{x})$ where $\mathbf{x} \sim \mathcal{D}$. Notice that since $\mathcal{D}$ is the uniform distribution, we have $\pi(\mathcal{D}) = \mathcal{D}$. From all the above, for every permutation $\pi$ with $\pi(j) = j$ we have:

$$-\frac{\partial}{\partial \mathbf{w}_j^{(i)}} L_{\chi_{\pi(I)},\mathcal{D}}(h_{\mathbf{u},\mathbf{w},\mathbf{b}}) = \mathbb{E}_{\mathbf{x} \sim \mathcal{D}} \left[ x_j u_i \sigma' \left( \left\langle \mathbf{w}^{(i)}, \mathbf{x} \right\rangle + b_i \right) \chi_{\pi(I)}(\mathbf{x}) \right]$$

$$= \mathbb{E}_{\mathbf{x} \sim \pi(\mathcal{D})} \left[ x_j u_i \sigma' \left( \left\langle \mathbf{w}^{(i)}, \pi^{-1}(\mathbf{x}) \right\rangle + b_i \right) \chi_I(\mathbf{x}) \right]$$

$$= \mathbb{E}_{\mathbf{x} \sim \mathcal{D}} \left[ x_j u_i \sigma' \left( \left\langle \pi(\mathbf{w}^{(i)}), \mathbf{x} \right\rangle + b_i \right) \chi_I(\mathbf{x}) \right] = -\frac{\partial}{\partial \mathbf{w}_j^{(i)}} L_{\chi_I,\mathcal{D}}(\pi(h_{\mathbf{u},\mathbf{w},\mathbf{b}}))$$

Fix some $I \subseteq [n]$ with $|I| = k$ and $j \in [n]$. Now, let $S_j$ be a set of permutations satisfying:

1. For all $\pi_1, \pi_2 \in S_j$ with $\pi_1 \neq \pi_2$ we have $\pi_1(I) \neq \pi_2(I)$.

2. For all $\pi \in S_j$ we have $\pi(j) = j$.

Note that if $j \notin I$ then the maximal size of such $S_j$ is $\binom{n-1}{k}$, and if $j \in I$ then the maximal size is $\binom{n-1}{k-1}$. Denote $g_j(\mathbf{x}) = x_j u_i \sigma'(\langle \mathbf{w}^{(i)}, \mathbf{x} \rangle + b_i)$. We denote the inner-product $\langle \psi, \phi \rangle_{\mathcal{D}} = \mathbb{E}_{\mathbf{x} \sim \mathcal{D}} [\psi(\mathbf{x})\phi(\mathbf{x})]$ and the induced norm $\|\psi\|_{\mathcal{D}} = \sqrt{\langle \psi, \psi \rangle_{\mathcal{D}}}$. Since $\{\chi_{I'}\}_{I' \subseteq [n]}$ is an orthonormal basis w.r.p. to $\langle \cdot, \cdot \rangle_{\mathcal{D}}$ from Parseval's equality we have:

$$\sum_{\pi \in S_j} \left( \frac{\partial}{\partial \mathbf{w}_j^{(i)}} L_{\chi_I,\mathcal{D}}(\pi(h_{\mathbf{u},\mathbf{w},\mathbf{b}})) \right)^2 = \sum_{\pi \in S} \left( \frac{\partial}{\partial \mathbf{w}_j^{(i)}} L_{\chi_{\pi(I)},\mathcal{D}}(h_{\mathbf{u},\mathbf{w},\mathbf{b}}) \right)^2$$

$$= \sum_{\pi \in S} \left\langle g_j, \chi_{\pi(I)} \right\rangle_{\mathcal{D}}^2 \leq \sum_{I' \subseteq [n]} \left\langle g_j, \chi_{I'} \right\rangle_{\mathcal{D}}^2 = \|g_j\|_{\mathcal{D}}^2 \leq 1$$

So, from the above we get that, taking $S_j$ of maximal size:

$$\mathbb{E}_{\pi \sim S_j} \left( \frac{\partial}{\partial \mathbf{w}_j^{(i)}} L_{\chi_I,\mathcal{D}}(\pi(h_{\mathbf{u},\mathbf{w},\mathbf{b}})) \right)^2 \leq |S_j|^{-1} \leq \min \left\{ \binom{n-1}{k}^{-1}, \binom{n-1}{k-1}^{-1} \right\}$$

Now, for some permutation invariant distribution of weights $\mathcal{W}$ we have:

$$\mathbb{E}_{\mathbf{w} \sim \mathcal{W}} \left( \frac{\partial}{\partial \mathbf{w}_j^{(i)}} L_{\chi_I,\mathcal{D}}(h_{\mathbf{u},\mathbf{w},\mathbf{b}}) \right)^2 = \mathbb{E}_{\mathbf{w} \sim \mathcal{W}} \mathbb{E}_{\pi \sim S_j} \left( \frac{\partial}{\partial \mathbf{w}_j^{(i)}} L_{\chi_I,\mathcal{D}}(\pi(h_{\mathbf{u},\mathbf{w},\mathbf{b}})) \right)^2 \leq |S_j|^{-1}$$

Summing over all neurons we get:

$$\mathbb{E}_{\mathbf{w} \sim \mathcal{W}} \left\| \frac{\partial}{\partial W} L_{\chi_I,\mathcal{D}}(h_{\mathbf{u},\mathbf{w},\mathbf{b}}) \right\|_2^2 \leq qn \cdot \min \left\{ \binom{n-1}{k}^{-1}, \binom{n-1}{k-1}^{-1} \right\}$$

We can use a similar argument to bound the gradient of $\mathbf{u}$. We leave the details to the appendix. $\square$

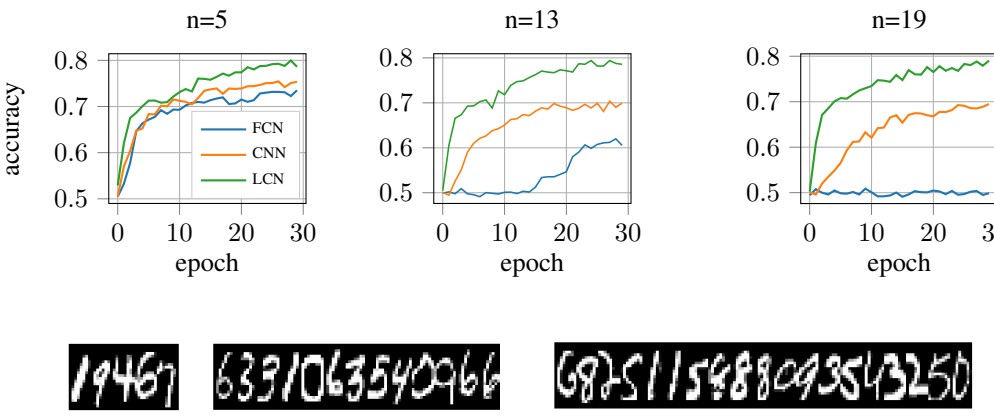

Figure 3: Top: Performance of different architectures on a size-$n$ MNIST sequences, where the label is determined by the parity of the central 3 digits. Bottom: MNIST sequences of varying length.

## 5 NEURAL ARCHITECTURE SEARCH

So far, we showed that while the $(\log n)$-pattern problem can be solved efficiently using a CNN, this problem is hard for a FCN to solve. Since the CNN architecture is designed for processing consecutive patterns of the inputs, it can easily find the pattern that determines the label. The FCN, however, disregards the order of the input bits, and so it cannot enjoy from the fact that the bits which determine the label are consecutive. In other words, the FCN architecture needs to learn the order of the bits, while the CNN already encodes this order in the architecture.

So, a FCN fails to recover the $k$-pattern since it does not assume anything about the order of the input bits. But, is it be possible to recover the order of the bits prior to training the network? Can we apply some algorithm that searches for an optimal architecture to solve the $k$-pattern problem? Such motivation stands behind the thriving research field of Neural Architecture Search algorithms (see Elsken et al. (2018) for a survey).

Unfortunately, we claim that if the order of the bits is not known to the learner, *no architecture search algorithm can help in solving the $k$-pattern problem*. To see this, it is enough to observe that when the order of the bits is unknown, the $k$-pattern problem is equivalent to the $k$-Junta problem: learning a function that depends on an arbitrary (not necessarily consecutive) set of $k$ bits from the input. Learning $k$-Juntas is a well-studied problem in the literature of learning theory (e.g., Mossel et al. (2003)). The best algorithm for solving the $(\log n)$-Junta problem runs in time $n^{O(\log n)}$, and no poly-time algorithm is known for solving this problem. Moreover, if we consider statistical-query algorithms (a wide family of algorithms, that only have access to estimations of query function on the distribution, e.g. Blum et al. (2003)), then existing lower bounds show that the $(\log n)$-Junta problem **cannot** be solved in polynomial time (Blum et al., 1994).

## 6 EXPERIMENTS

In the previous sections we showed a simplistic learning problem that can be solved using CNNs and LCNs, but is hard to solve using FCNs. In this problem, the label is determined by a few *consecutive* bits of the input. In this section we show some experiments that validate our theoretical results. In these experiments, the input to the network is a sequence of $n$ MNIST digits, where each digit is scaled and cropped to a size of $24 \times 8$. We then train three different network architectures: FCN, CNN and LCN. The CNN and LCN architectures have kernels of size $24 \times 24$, so that 3 MNIST digits fit in a single kernel. In all the architectures we use a single hidden-layer with 1024 neurons, and ReLU activation. The networks are trained with AdaDelta optimizer for 30 epochs [1].

---

[1] In each epoch we randomly shuffle the sequence of the digits.

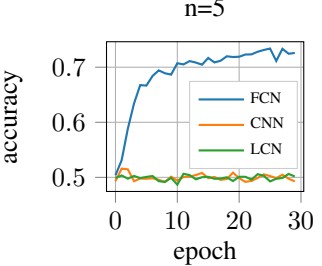 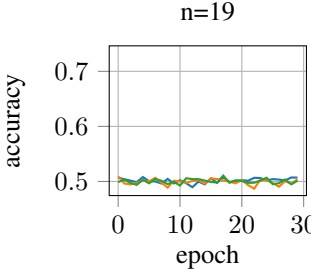

Figure 4: $n$-sequence MNIST with non-consecutive parity.

In the first experiment, the label of the example is set to be the parity of the sum of the 3 consecutive digits located in the middle of the sequence. So, as in our theoretical analysis, the label is determined by a small area of consecutive bits of the input. Figure 3 shows the results of this experiment. As can be clearly seen, the CNN and LCN architectures achieve good performance regardless of the choice of $n$, where the performance of the FCN architectures critically degrades for larger $n$, achieving only chance-level performance when $n = 19$. We also observe that LCN has a clear advantage over CNN in this task. As noted, our primary focus is on demonstrating the superiority of locality-based architectures, such as CNN and LCN, and we leave the comparison between the two to future work.

Our second experiment is very similar to the first, but instead of taking the label to be the parity of 3 consecutive digits, we calculate the label based on 3 digits that are far apart. Namely, we take the parity of the first, middle and last digits of the sequence. The results of this experiment are shown in Figure 4. As can be seen, for small $n$, FCN performs much better than CNN and LCN. This demonstrates that when we break the local structure, the advantage of CNN and LCN disappears, and using FCN becomes a better choice. However, for large $n$, all architectures perform poorly.

**Acknowledgements:** This research is supported by the European Research Council (TheoryDL project). We thank Tomaso Poggio for raising the main question tackled in this paper and for valuable discussion and comments

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
