# OpenReview forum: "Computational Separation Between Convolutional and Fully-Connected Networks"
_ICLR.cc/2021/Conference — ICLR 2021 Poster_

### Official Review · AnonReviewer1 · 2020-10-17
**A good paper showing a computational separation between CNN and FC.**

**Rating:** 8
**Confidence:** 3

**Review:**

The paper establishes a separation result between a convolutional network (CNN) and the fully connected network (FC).
Specifically, the authors show that a polynomial-size of CNN trained by gradient descent with a polynomial number of iterations can learn functions that depend only on a small pattern of $k$-consecutive bits of the input,
where $k = O(\log n)$ and $n$ is the input dimension. On the other hand, they show that for FC, gradient descent fails to learn a $k=O(\log n)$-parity function unless the network size is $\Omega( n^{\log n})$. Hence, the separation result in the computational aspect is proven. An experiment is conducted to support the theoretical results.


I believe this paper makes a significant contribution. The writing is good and the proof is short and concise. The authors first discuss other possible explanations for the observation that CNN performs better than FC in practice and argue that parameter efficiency and weight sharing might not be the main reason for the superiority of CNN. Then, in the rest of the paper, the authors show that the key is that CNN is able to exploit the locality in the data.


Q: It appears that in the analysis of the positive result regarding CNN, the training is in the neural tangent kernel regime, as you have to bound the deviation from the initial point and that you show a randomly initialized CNN already works well for the $k$-pattern problem. Does it suggest that the kernel method can learn the $k$-parity function? What is the catch here? I saw the sparsity here but I am hoping the authors can explain the intuition behind their results more.

---

> ### Author Response · Authors · 2020-11-14
> **Response to Reviewer #1**
>
> Thank you for the very positive review and encouraging feedback.
> As for your question, our result shows that the k-pattern problem (i.e., a parity of **consecutive** bits), is learnable using a CNN. This analysis indeed uses the fact that the kernel induced by the CNN architecture is suitable for learning parities of consecutive bits. However, the fact that the bits are consecutive plays a key role here, and this does not imply that kernels can learn general k-parities.

---

### Official Review · AnonReviewer2 · 2020-10-18
**Review for separation CNN and FCNN**

**Rating:** 8
**Confidence:** 3

**Review:**

It is well-known that neural networks (NN) perform very well in various areas and in particular if one looks at computer vision convolutional neural networks perform very well. Although convolutional neural networks (CNN) are limited in their architecture (since they only allow nearest-neighbour connections) compared to fully-connected NNs (FCNN), their superiority in performance is unclear. In this paper they answer the following fundamental question: can one formally show that CNNs are better than FCNNs for a specific learning task? In this direction they answer in the affirmative.  In particular, more than just giving an example, they show that an interesting property called locality, instead of other parameters like parameter and efficiency weight sharing is the reason for its superior performance.

In order to exhibit their separation, they consider the class which they denote k-patterns which is is simply a k-junta, i.e., f is a function on n-bits but when defining f it only depends on an arbitrary function of the k out of the n inputs bits, which are fixed when defining f, but unknown to a learner. In this paper, the authors consider such k-patterns which are very local simply by definition and show that for these class of functions, training a CNN with  gradient descent can learn these functions in polynomial time (for k = O(log n)) however gradient descent fails to learn these functions when training a FCNN. The proof of the lower bound for FCNN uses the well-known statistical query framework which shows that using gradient descent to learn k-parities requires time n^k (and in their application n^ log n). In order to prove the upper bound for CNN, they combine various techniques that were introduced in many recent works (in a non-trivial way).

Overall, I think this paper resolves an important and interesting question. Given that most of machine learning is well understood in the non-rigorous setting, giving a rigorous separation between CNN and FCNN is an important problem and they have resolved it. If I had to be nit-picky, I would say the concept class they give this separation for is well-suited for their separation and doesn't really say much more than that, but I think it's already interesting that one can show such a separation theoretically.

---

> ### Author Response · Authors · 2020-11-14
> **Response to Reviewer #2**
>
> Thank you for the very positive review and encouraging feedback!

---

### Official Review · AnonReviewer3 · 2020-10-28

**Rating:** 6
**Confidence:** 4

**Review:**

Summary:

The paper studies the problem of computational separation between two-layer convolutional neural network (CNN) and fully-connected neural network (FCN). It shows that there is a class of function, which is defined in the paper as k-pattern, such that CNN could learn this class within polynomial time with polynomial number of neurons while FCN needs exponential number of neurons to avoid getting stuck at initialization.

Pros:
- The problem about the separation between neural network architectures, such as CNN and FCN in this paper, is an interesting and important topic. This paper gives evidences that CNN is preferred when input data has local structure.
- The result theoretically justifies the intuition that CNN could utilize locality of data.
- Experiments are also provided to support the results.
- The paper overall is well motivated and easy to follow.


Cons/Questions: I think overall this paper is good, below are some of my minor concerns/questions.
- The current convergence proof relies on the online optimization technique, which could show the average loss across all iteration is small. I admit that the setting is different from NTK analysis. However, given that current proof also relies on the fact that the first layer does not move to much from initialization, which is quite similar with the idea of NTK analysis, I was wondering if some techniques in NTK analysis could be applied to show the convergence of last iteration.
- As mentioned in the paper, the analysis for CNN uses a non-standard initialization.
- The result for FCN shows that the gradient at initialization is small in expectation. I was wondering if a high probability result could also hold. Also, to show gradient descent will get stuck at initialization, seems author(s) implicitly assume that step size is upper bounded. Otherwise, one can take large enough step size to escape the initialization. It would be better to explicitly mention this.


Minor:
- Notations for u at time T seems to be inconsistent. In Theorem 4, it is u^{(0,j)}, while in Lemma 8 it becomes u^{(j,T)}.
- Proof. of Theorem x. -> Proof of Theorem x.

After authors' response:
Thanks for the response! I will keep my score.

---

> ### Author Response · Authors · 2020-11-14
> **Response to Reviewer #3**
>
> Thank you for your positive review.
> Regarding your questions and comments:
> - Indeed, the convergence result is in a similar spirit as the NTK analysis. We chose a slightly different approach for the convergence analysis than the one that is used in most NTK works, but as you mentioned this is very much aligned with the idea of NTK analysis. We believe a similar result can be shown via "standard" NTK analysis.
> - We assume a non-standard initialization as it greatly simplifies the analysis, but we believe a very similar result can be shown with standard initialization (for example, via NTK analysis). However, we chose to show the simpler proof to make the paper more readable.
> - You are right, we do need to assume that the step size is upper bounded by some constant (in fact, we can even allow it to grow polynomially with the dimension). We will mention this explicitly in the paper. Deriving a high-probability result in this case is straightforward, for example using Markov's inequality.
>
> We will additionally fix the minor comments.

---

### Official Review · AnonReviewer4 · 2020-10-30
**Interesting attempt but the argument seems incomplete.**

**Rating:** 5
**Confidence:** 4

**Review:**

This paper tries to provide an explanation for the performance advantage of CNNs over FCNs  by trying to identify labelling functions where the performance gap between the two for supervised learning can be proven rigorously. The identify an interesting class of such labelling functions : the pseudo-Boolean functions called the k-pattern functions and they are able to show that depth 2 CNNs can easily learn this class.  I am personally very excited by Theorem 4. I think this is a very cool observation and this could lead to a lot of future work.  But unfortunately this is where the concrete part of the argument in this paper stops.

I dont see why the argument in Section 4 is enough to prove that this k-pattern is hard to learn for FCNs. To the best of my understanding Section 4 is not an impossibility result or a lowerbound to learning but is only making a plausibility argument towards such. I think Theorem 12 is essentially motivating the conjecture that if the expected gradient at initialization can be arbitrarily small then maybe GD will find it hard to learn. But this conjecture is not getting proven here.

I think this point needs to be made clear and we need something stronger to be said about this for Section 4 to qualify as a serious piece of argument. Currently I can think of "counter examples" which I would like the authors to address :  Consider the ODE : dx/dt = (2/t^3)*e^{-1/t^2} with the initial condition x(0) = 0. As you can convince yourself dx/dt at t=0 = 0 So this gradient flow starts with exactly 0 velocity/gradient but this ODE does integrate to x(t) = e^{-1/t^2} which is an increasing function of time and it gets further far away from the origin as time passes.

So if this phenomenon can happen with ODEs why cant this happen with GDs - or even SGDs where the gradient oracle is say initially sending noisy estimates of this dx/dt above and one is initializing at the origin. There noise should maybe only help rather than hurt. So given this I am not convinced that having the expected gradient at initialization being low is a barrier to GD's or SGD's movement.  I would like to know if there is a reason why my intuition based on this example could be wrong!

Also the notation "x_{j,..j+k}" seems to suggest that this vector comprises of the entries of the vector of x at coordinates j,j+1,..,j+k. So this is a "k+1" dimensional vector.  Then shouldn't W be a q x (k+1) dimensional vector and similar adjustments be made everywhere else?

---

> ### Author Response · Authors · 2020-11-14
> **Response to Reviewer #4**
>
> Thank you for the review.
> Regarding your major concern, as to why bounding the average gradient norm at a random point is sufficient for showing that gradient-descent fails to learn the target function: First, observe that the upper bound on the gradient norm decays super-polynomially with the dimension (i.e., at a rate of n^(-log n) when k = log n). This means that with overwhelming probability over a random choice of the network parameters, the gradient is extremely small. In fact, if we run GD on a finite-precision machine (as we typically do), the true population gradient is effectively zero. Even if we assume that we run GD with stochastic noise, due to the previous observation doing so is equivalent to sampling randomly independent points around the initialization (as the path of GD is completely governed by the random noise). Therefore, if we run noisy GD for a polynomial number of iterations, it will return a function that is independent from the target function, and thus the optimization will fail.
> This argument has been shown formally in various previous works (see for example https://arxiv.org/pdf/1609.01037.pdf, https://arxiv.org/abs/1812.06369, https://arxiv.org/abs/2008.08059), where it has been shown how such bound on the gradient norm can be translated to an argument about the failure of different variants of GD. We will update our paper with a more elaborate version of this argument.
> As for the other (minor) comment, you are right, the notation x_{j ... j+k} is indeed confusing, we will fix this accordingly.

---

### Decision · Program_Chairs · 2021-01-07
**Final Decision**

**Decision:**

Accept (Poster)

**Comment:**

This paper aims at answering an interesting question that puzzles the whole community of deep learning: why CNNs perform better than FCNs? The authors show that CNNs can solve the k-pattern problem much more efficiently than FCNs, which partially contributes to the answer of the question.

Pros:
1. Studies an interesting question on DNNs.
2. Constructs a specific problem, the k-pattern problem, so that CNNs can solve much more efficiently than FCNs.

Cons:
1. The analysis is only a very limited answer to the question. It only shows that CNNs are more efficient than FCNs on a very specific problem, which is of little interest to the community. On the one hand, people want to see the advantage of CNNs on more common problems, perhaps the image recognition problem (The AC understands that analyzing this problem is nearly impossible. It is just for hinting the choice of problems to analyze)? On the other hand, maybe others can find another specific problem that FCNs can solve much more efficiently than CNNs. If so, the value of this paper will be totally gone. The authors did not exclude such a possibility (Nonetheless, it is still a computational "separation" between CNNs and FCNs :)).
2. Reviewer #4 pointed out an issue in the proof. The response from the authors, though looked promising, did not fully convince the reviewer (in the confidential comment). Reviewer #3 also raised a question on the bounded stepsize. The authors should address both issues.

Overall, since the problem studied is of great interest to the community and the analysis is mostly sound, the AC recommended acceptance.